# Minimally Invasive One-Docking, Two-Target, and Three-Port Robotic-Assisted Nephroureterectomy: Redefining Surgical Approach

**DOI:** 10.3390/cancers17040627

**Published:** 2025-02-13

**Authors:** Yarden Zohar, Ben Hefer, Itay Vazana, Muhammad H. Jabareen, Rabea Moed, Elad Mazor, Emilio Charabati, Nimer Alsaraia, Nicola J. Mabjeesh

**Affiliations:** Department of Urology, Soroka University Medical Center, Faculty of Health Science, Ben-Gurion University of Negev, P.O. Box 151, Beer Sheva 84101, Israel; yardenzo@clalit.org.il (Y.Z.);

**Keywords:** da Vinci robotic system, multi-quadrant minimally invasive approach, nephroureterectomy, robotic single docking, three ports, robotic-assisted laparoscopic radical nephroureterectomy, robotic retargeting

## Abstract

This study focuses on robotic radical nephroureterectomy performed for patients with urothelial carcinoma in their kidneys and ureters. The da Vinci Xi platform, which enhances minimally invasive surgical techniques was applied. We aimed to optimize the surgical approach by reducing the number of ports used, and improving visualization during the procedure. The study involved 15 patients operated on by a single surgeon between 2019 and 2024, using a one docking, three-port technique. The median operative time was 133 min, and most surgeries were completed within 150 min. Blood loss was minimal, and complication rate was low. In conclusion, the study highlights that the three-port technique combined with the Xi platform is a safe and effective method for performing robotic radical nephroureterectomy, facilitating quicker procedures and achieving surgical goals.

## 1. Introduction

Radical nephroureterectomy (NU) with bladder cuff excision (BCE) is the gold-standard treatment for upper tract urothelial carcinoma (UTUC). Current guidelines indicate the importance of the surgical management of UTUC; however, there is no consensus regarding the operative approach [1]. The ‘master–slave’ robotic system was first introduced in the late 1990s, and the first robotic-assisted laparoscopic urology surgery was performed in 2000 [2]. Since then, the da Vinci platform has evolved. In general, the da Vinci robotic platform is superior to the laparoscopic technique owing to its 3D imaging, and the wristed configuration provides a high degree of freedom, allowing accurate and precise movements. Despite these advantages, the da Vinci platform faces limitations, including its high financial cost and economic burden, which restrict its accessibility in many public health institutions worldwide [3].

Although robot-assisted laparoscopic radical NU has already been described, its intraoperative management remains highly versatile and largely depends on surgeon experience and expertise. Standardized radical NU tetrafecta includes negative soft tissue surgical margins, BCE, lymph node dissection (LND) (for high-risk, non-metastatic disease), and no evidence of recurrence within 12 months [4,5]. The key operative goals are to minimize the overall operative time, prevent tumor seeding, and reduce the risk of urine leakage. However, the extensive nature of this surgery, especially when LND is performed, places patients at an increased risk of complications, such as major organ injury, bowel perforation, and significant blood loss. The da Vinci Xi robotic system offers distinct advantages in such complex procedures, particularly in its ability to facilitate multi-quadrant minimally invasive surgery [6]. This article presents our experience with 15 patients who underwent multi-quadrant robotic radical nephroureterectomy (RRNU) using the Xi platform. By utilizing only three ports and the camera-hop feature, we redefined the single-docking RRNU procedure to streamline the procedure and achieve favorable outcomes.

## 2. Patients and Methods

With the approval of the Institutional Ethics Committee, we retrospectively collected data from patients diagnosed with UTUC who underwent RRNU between 2019 and 2024. Patients who underwent NU for indications other than UTUC were excluded from the study. All patients included in the study had a suspicious lesion on imaging and were diagnosed with UTUC through diagnostic ureteroscopy (Dx URS). Following a multidisciplinary evaluation of all associated risks, they were referred for radical NU (Figure 1). Notably, updated guidelines now allow a radical surgical approach based solely on imaging or visual appearance, without requiring biopsy-confirmed malignancy [4]. However, our study was conducted before these guideline revisions. The data included preoperative, intraoperative, and postoperative surgical details, with a follow-up period of 12 months to monitor the outcomes. All the procedures were performed by a single surgeon with up to 10 years of experience in robotic surgery. Fifteen patients met the inclusion criteria and were included in this study.

## 3. Surgical Intervention

All 15 patients underwent RRNU with BCE by using the Xi da Vinci robotic system. The operator employed a unique intraoperative technique based on single docking for multi-quadrant surgery. After catheter insertion, patients were placed in a modified lateral decubitus position with 30-degree flexion. Trocar placement was planned along the paramedian line, with 8 cm spacing between the trocars. Three 8 mm trocars were used to dock the robotic arms, and a robotic camera was inserted through the middle port. An additional 12 mm trocar was placed in the abdominal midline, inferior to the umbilicus, and 5 cm from both the camera and the third arm trocars, forming a triangular configuration (Figure 1). The camera was initially directed toward the renal area for targeting, enabling optimal robotic arm positioning and rotation for the nephrectomy portion of the procedure.

Once docking was completed, the surgical procedure began with retroperitoneal exposure, facilitated by the dissection of Toldt’s white line. After exposing the kidney, pre-renal tissue was carefully dissected and separated at the superior lateral border. After stapling the main renal blood vessels, caudal dissection was performed at the mid-ureteral level, approximately at the L2–L3 level, parallel to the gonadal vessels. Next, the assistant disconnected the robotic endpoint instruments and redirected the camera toward the pelvis. This step allowed for the retargeting of the robotic camera and reconfiguration of the robotic arms to face the pelvic area. Once the endpoint instruments were reconnected, dissection of the distal ureter was completed, and extravesical BCE was performed.

A Hem-o-lok clip was placed on the distal ureter (depending on the location of the tumor) before complete transection and separation. An inverted U-shaped stitch using a 3-0 V-Loc suture was applied at the ureteral margin to prevent urine leakage during the excision. Following the completion of the BCE procedure, the specimen was placed directly into a sealed laparoscopic bag, and the bladder was sutured with 3-0 V-Loc. In selected cases, LND was performed along the template, extending from the ipsilateral renal vein to the common iliac vessels including the external and internal iliac vessels. To minimize spillage, a separate endo-bag was used for LND specimens. A Jackson–Pratt (JP) drain was placed along the retroperitoneum from the resection bed of the kidney to the pelvis.

## 4. Results

Fifteen patients were included in the study, with data collected between 2019 and 2024 with a minimal follow-up period of 12 months, except for the last patient who died postoperatively. Table 1 summarizes the patients’ demographic characteristics. The preoperative variables are summarized in Table 2. The median patient age was 79 years with an average age-adjusted Charlson comorbidity index score of 8.

### 4.1. Operative Details

The median total operative time was 133 min, with 60% of the procedures completed within 150 min. Notably, LND did not significantly prolong the operative time. The median estimated blood loss (EBL) was 100 mL, with two patients requiring intraoperative blood transfusions despite controlled bleeding, attributed to their low preoperative hemoglobin levels. None of the cases required surgical technique conversion for the RRNU portion.

### 4.2. Lymph Node Dissection and Outcomes

LND was performed in six patients, with two showing positive nodes. Despite the extent of the procedure, no major intra- or postoperative complications were noted.

### 4.3. Postoperative Outcomes (Table 3)

*Complications*: The overall major complication rate (Clavien-Dindo grade > III) was 13.33%. One patient succumbed to cardiac arrest on postoperative day (POD) 2. Resuscitation was initiated immediately, followed by an emergency coronary angiography. Unfortunately, the last patient developed irreversible cardiac arrhythmias and did not respond to further resuscitation. The second patient was readmitted on POD 7 with type II myocardial infarction and required angioembolization.

**Table 3 cancers-17-00627-t003:** Postoperative variables.

Postoperative Lab Values	(N = 15)
**Hb (g/d** **L)**	
Mean	11.2
Median	11
Range	7.5–13.9
**Creatinine**	
Mean	1.6
Median	1.5
Range	1–3.7
**eGFR (mL/min/1.73 m^2^)**	
Mean	48
Median	45
Range	15–82
**Postoperative hospitalization**	**(N = 14)**
**Length of stay (days)**	
Mean	5.8
Median	5
Range	3–10 months
**Day of catheter withdrawal**	
Mean	15.5
Median	4.5
Range	3–150 *
**Day of drain withdrawal**	
Mean	4
Median	3
Range	2–14 months
**Complications rated by Clavien-Dindo Score (N = 15)**	
0–1	7 (46.6%)
2–3 months	6 (40%)
4	1 (6.66%)
5	1 (6.66%)
**Complications noted during postoperative period (N = 15)**	
Fever	3
Urosepsis	2
Blood transfusion	5
Atelectasis	3
Surgical site infection	1
Ileus	2
Acute kidney injury	9
Type II MI	1
Sudden cardiac arrest	1
ICU Observation	1

* One patient remained with indwelling catheter until prostatectomy (TUR-P).

*Renal Function*: The median postoperative estimated glomerular filtration rate (eGFR) was 45 mL/min/1.73 m^2^, representing an average decline of 21.87 mL/min/1.73 m^2^. Three patients with preoperative renal failure developed chronic renal failure (CRF). Among six patients who developed acute kidney injury postoperatively, all maintained an eGFR ≥ 50 mL/min/1.73 m^2^ at one month.

*Hospital Stay*: The average hospital stay was five days. Catheters were removed on POD 3 to 5, except in one patient with bladder outlet obstruction who later underwent prostate surgery. Drains were removed on POD 4.

### 4.4. Oncological Outcomes

A period of 12 months was defined as the minimum timeframe for follow-up. This period was characterized by close monitoring; thus, the majority of metastatic or recurrence cases were identified within this timeframe. Although international guidelines indicate that close follow-up has limited evidence for both high- and low-risk disease following radical NU, our institution continues to adhere to this practice, ensuring diligent monitoring [1]. At 12 months, the overall survival and disease-free survival rates were 78.57% and 42.86%, respectively. Of the two patients who received neoadjuvant therapy, one achieved a pathological complete response (pT0) with no evidence of disease (NED) in the 12-month period. Among the six patients who completed adjuvant therapy, two achieved a complete response. Five patients presented with metastatic disease involving the liver, retroperitoneal lymph nodes, lungs, or bones. Among these, three died from the disease within an average of eight months postoperatively. Two patients continued the second-line adjuvant therapy with avelumab. As shown in Table 4, three patients presented with non-urothelial metastatic disease, with recurrence occurring beyond the 12-month follow-up period defined as our study’s follow-up timeframe.

*Bladder Recurrence*: Nine patients resumed cystoscopy follow-up; of these, two progressed to metastatic disease. Among the seven non-metastatic patients, three developed bladder recurrence at an average of six months postoperatively, requiring transurethral resection (TURBT) and continued Bacillus Calmette–Guérin (BCG) therapy. One patient refused to undergo BCG instillations.

### 4.5. Sub-Group Analysis

*With regard to Charlson adjusted comorbidity index (CCI)*: It is important to note that the cohort’s population had a median CCI score of 8. To assess whether this technique is suitable for highly morbid patients, we conducted a sub-analysis using a CCI cut-off point of ≥5. A total of 11 out of the 15 patients met this criterion. Among them, two patients died from metastatic disease, one died from cardiac arrest as mentioned, and two patients are still alive, presenting with metastatic UTUC and receiving adjuvant therapy. The overall survival rate in this sub-analysis group was 72.72%, and the disease-free survival rate was 54.54%.

*With regard to age*: Regarding age, the cohort’s median age was 79 years. We conducted a sub-group analysis to assess whether the elderly population would benefit from the procedure, using 80 years as the cut-off. A total of seven patients were 80 years or older. Of them, only one patient died from metastatic disease. The remaining six patients survived with no evidence of disease at the 12-month follow-up. The overall survival rate in this subgroup was 85.71%, with the same rate of disease-free survival.

## 5. Discussion

Early publications on radical NU involved a combination of laparoscopic nephrectomy and robotic distal ureterectomy with BCE [7]. With technological advancements and increased surgical experience, the robotic approach has become the preferred method for radical NU [8]. However, one of the major drawbacks of RRNU compared to open or laparoscopic approaches is its high cost and time consumption [9]. Despite the minimally invasive nature of the procedure, the prolonged operative time limits its advantages in terms of complication rate [10]. Therefore, optimizing even minor factors to reduce procedure time is essential when performing robotic surgery.

Single-docking robotic NU has been described in the literature, although there is no consensus on the optimal number of ports, their placement, or step-by-step procedural methods. Recent advancements have focused on utilizing the camera-hop technique of the Xi robotic system, which involves intraoperative reconfiguration from the cephalad to caudal direction. Although this approach has been shown to be safe, it has not been universally adopted [11,12,13,14].

In our cohort, we employed a relatively under-reported technique for RRNU using only three robotic arm ports and one assistant port: two 8 mm ports for robotic endpoint instruments, one 8 mm port for the camera, and one 12 mm AirSeal assistant port (Figure 1). This minimalist approach allowed for efficient multi-quadrant dissection while minimizing port placement, reducing arm collision, and shortening the overall operative time, thus facilitating a more focused and streamlined surgical process. The intra- and postoperative complications are summarized in Table 5 and Table 6, respectively.

The median operative time for our cohort was 133 min, which is significantly lower than the median operative time of 203 min reported by Pathak et al. in their 90-patient cohort using a four-port RRNU technique [13]. Additionally, intraoperative complications in our cohort were minimal, with only two patients requiring blood transfusions, both with controlled bleeding. Importantly, none of the patients required conversion of the surgical approach due to bowel complications. The postoperative major complication rate, defined as Clavien-Dindo grade > III, was 13.33%, which included one death from sudden cardiac arrest and one patient with type II myocardial infarction. These complications should be considered in the context of our patient population, which primarily consisted of patients with a high burden of comorbidities, as reflected by a median CCI of 8 (Table 1). The CCI is designed to preoperatively identify patients at greater risk of intra- and postoperative complications [15]. In our analysis of the subgroup of patients with a CCI ≥ 5, we observed a total complication rate of 18%.

The inclusion of octogenarian patients in the surgical management of UTUC has been a frequent focus of previous studies. Large cohorts have demonstrated the benefits of minimally invasive RRNU in elderly populations, emphasizing its low complication rates and relatively high overall survival rates [16,17,18].

In our sub-analytical group of patients aged over 80 years, this study corroborates earlier findings, confirming the safety and feasibility of minimally invasive NU. Within this group, we observed a relatively high survival rate of 85.7% and an identical disease remission rate, with a mean follow-up period of 18 months. The complication rate in this subgroup was 14%, comparable to the overall study population.

A review of the literature reveals diverse opinions regarding the factors that most significantly influence surgical outcomes. Wallace et al. reported higher mortality rates among elderly patients; however, their study encompassed 12 different urological procedures rather than focusing solely on NU [19]. In contrast, data from 2,352 patients in the CROES-UTUC registry demonstrated survival benefits associated with radical NU, regardless of age or comorbidities [17].

Our findings suggest that a patient’s morbidity profile, as reflected by the CCI, may have a more significant impact on complication rates than age alone [15]. Nonetheless, further prospective studies are necessary to validate these findings.

It is worth noting, however, that the majority of clinical research supports the superiority of radical NU for UTUC, even in the octogenarian population [16,17,20,21]. Multiple clinical trials have compared minimally invasive approaches to open surgery, all indicating the superiority of the minimally invasive approach, particularly robotic surgery [18,21,22,23,24]. Unfortunately, one of the drawbacks of the robotic approach, as mentioned, is its time consumption, which may limit its use in the fragile elderly population, further underscoring the advantage of our time-efficient surgical technique [25,26].

While several cohorts have described the use of the camera-hop technique with the da Vinci Xi robotic system for NU, to the best of our knowledge, no study has utilized the three-port single-docking technique described in this report. Studies by Nanigian et al. (2006), Lee et al. (2013), Patel et al. (2015), Pathak et al. (2020), and others have reported various robotic approaches; however, they did not employ this specific configuration [9,13,27,28]. Pathak et al. utilized a four-port technique for RRNU and noted several intraoperative complications, including duodenal injuries, emphasizing the advantage of a reduced-port technique in minimizing the risk of such injuries. Additionally, they suggested that the integration of a movable table could further aid in dissection and prevent arm collisions. Arm collision is another challenge that our three-port technique effectively mitigates, particularly when combined with the Xi arm adjustment switch button [13].

One of the largest studies on RRNU, the ROBUUST cohort, analyzed 1118 patients and reported a median operative time of 212.5 min, along with a postoperative complication rate of 14.1%. While these results provide valuable insights, it is important to note that the ROBUUST study did not standardize robotic system use or surgical techniques, as it included data from multiple centers with varying levels of surgeon expertise. This variability makes direct comparison with our cohort challenging [29]. However, when comparing our results to a more homogeneous group—specifically, the study by Veccia and colleagues published in 2022, which analyzed 148 patients from the ROBUUST cohort who underwent single-stage RRNU with the Xi robotic system—the results were more directly comparable [30]. Their study reported a median operative time of 215.5 min (range: 160.5–290.0), with matched estimated blood loss (EBL) and major complication rates. In contrast, our cohort achieved a median operative time of 133 min, which was considerably shorter, while maintaining a similar complication profile. Although our cohort was small, the results align closely with those of the ROBUUST group in terms of overall complication rates and EBL [29]. Furthermore, our reduced operative time was a notable improvement compared with that of the ROBUUST group, underscoring the time efficiency of our three-port approach. These findings suggest that the three-port technique is not only safe, but also highly time-efficient for performing RRNU (Table 4).

Our study specifically excluded benign cases, focusing solely on malignant UTUC to better evaluate the oncological safety of the three-port single-stage robotic approach. Notably, all patients in our cohort had negative surgical margins, including BCE, and none experienced recurrence in the resection bed during the 12-month follow-up period, even among patients with metastasis (disregarding retroperitoneal lymph nodes). This highlights the oncological efficacy of our technique, which is comparable to that of larger and more complex cohorts.

Our study has several strengths. Uniformity in the surgical technique, patient positioning, trocar placement, intraoperative instruments, and stitches ensured consistency and reliability across all cases. The use of a transperitoneal approach and the Xi robotic program further minimized spillage and optimized procedural time, contributing to the overall efficiency of the technique. However, limitations exist, including the retrospective nature of the study, lack of console time data, and variability in non-robotic procedures such as cystoscopy and Mitomycin C (MMC) instillations. In addition, because the procedure was performed by a single surgeon, the results were subject to performance bias. On the other hand, the experience of a single surgeon enhances the reliability and accuracy of the results, strengthening the feasibility and safety of the three-port single-docking technique.

## 6. Conclusions

RRNU is a complex procedure that demands a high level of expertise, which can be enhanced through advancements in robotic technology, particularly the upgrade from the X to the Xi robotic platform. Our study demonstrated that the three-port, single-docking NU procedure is both feasible and safe, particularly when the retargeting capabilities of the Xi system are utilized. This cohort achieved the defined tetrafecta for NU, with results comparable to those of the largest published case series using four ports. Notably, the three-port, single-docking technique showed superior performance in terms of overall operative time. To further validate these findings, prospective comparisons between three- and four-port single-docking techniques using the Xi da Vinci program are required. Additionally, the integration of artificial intelligence to optimize camera movement and predict shifts in the surgical field may further improve the efficiency and precision of this approach.

## Data Availability

The datasets generated and/or analyzed during the current study are available from the corresponding author upon reasonable request.

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
