# Peer review of "Minimally Invasive One-Docking, Two-Target, and Three-Port Robotic-Assisted Nephroureterectomy: Redefining Surgical Approach"

_cancers, 2025, doi:10.3390/cancers17040627_

Round 1
Reviewer 1 Report
Comments and Suggestions for Authors
Thank you very much for the opportunity to read this manuscript. This study presents a refined technique for robotic radical nephroureterectomy (RRNU) using the da Vinci Xi platform. The approach utilizes a single docking, three-port technique, optimizing port usage and template visualization. The study analyzed 15 patients who underwent RRNU between 2019 and 2024. Results showed a median operative time of 162 minutes, with 60% of procedures completed within 150 minutes. Median blood loss was 100 ml, and lymph node dissection did not significantly impact overall operative time. The intraoperative and major postoperative complication rates were 13.3%. The authors conclude that this three-port technique with the Xi platform's camera-hop feature is safe and effective for RRNU, facilitating procedural goals and reducing operative time.
However, some points need revision.
1. Figure 1. please start with capital letter “Patient” instead “patient”. Moreover, the inscriptions on Figure 1 are unclear, e.g. 12mm post side ...
2. In table 1. Prostate / breast – 3? According to Table 6, only one patient with metastatic breast cancer and 2 with metastatic prostate cancer were reported. This raises questions regarding patient selection for this study – out of 15 patients, 1 succumbed to cardiac failure. Consequently, of the 14 patients analyzed at the 12-month follow-up, 3 presented with metastatic disease of the prostate and breast. What was the rationale for including these patients in the study?
The authors indicate in Table 2 that PET-CT was performed only in 4 patients – what were the indications for this procedure? Why was it not conducted on all patients? This inquiry pertains to potential bias in patient selection, considering the relatively small cohort of 15 patients over a 5-year study duration.
3. Why was only 12-month follow-up applied?
4. What is the indication for lymphadenectomy, please explain.
5. Considering the uncertainties mentioned in sub-items 2 and 4, please provide a comprehensive diagnostic protocol, exclusive of URS, detailing the specific tests that should be conducted on each patient deemed eligible for the procedure. (A flow chart may be the most appropriate format for presentation.)
Author Response
Author's Reply to the Review Report (Reviewer 1)
Dear Reviewer I,
We thank you for your quick and highly informative review. Thanks to your useful comments we have made changes as follows:
- Figure 1. please start with capital letter “Patient” instead “patient”. Moreover, the inscriptions on Figure 1 are unclear, e.g. 12mm post side ...
Author’s response: Was reviewed and changed
- In table 1. Prostate / breast – 3? According to Table 6, only one patient with metastatic breast cancer and 2 with metastatic prostate cancer were reported. This raises questions regarding patient selection for this study – out of 15 patients, 1 succumbed to cardiac failure. Consequently, of the 14 patients analyzed at the 12-month follow-up, 3 presented with metastatic disease of the prostate and breast. What was the rationale for including these patients in the study?
Author’s response: In Table 6, we noted that three patients presented with non-urothelial carcinoma (non-UC) metastatic disease during the follow-up period, which was defined as at least 12 months. All three patients were diagnosed with metastatic disease during the post-operative follow-up period, that in their case was longer than 24 months after their surgical procedure. Of note, their non-UC oncological disease was under control (defined by the oncologists as NED- no evidence of disease). Consequently, these cases could not be excluded as their condition was unknown at the time of inclusion. This observation highlights that, although these patients had active oncological diseases, they were not related to UC. This underscores the safety of the surgical technique, as it does not predispose patients to seeding or recurrence of upper tract urothelial carcinoma (UTUC).
We have added this note in Line 184-186:
“ . As shown in Table 6, three patients presented with non-urothelial metastatic disease, with recurrence occurring beyond the 12-month follow-up period defined as our study's follow-up timeframe.”
The authors indicate in Table 2 that PET-CT was performed only in 4 patients – what were the indications for this procedure? Why was it not conducted on all patients? This inquiry pertains to potential bias in patient selection, considering the relatively small cohort of 15 patients over a 5-year study duration.
Author’s response: For upper tract urothelial carcinoma (UTUC), PET-CT scans have limited evidence for diagnosis and staging and are therefore not routinely performed. However, in selected cases with bulky disease or a high suspicion of local invasion or metastatic disease, PET-CT may be performed prior to surgery. In certain cases, management of locally invasive disease may include neoadjuvant therapy prior to surgical intervention. These changes were implemented recently, following the updated algorithm published in April 2024 (Pandolfo et al., 2024).
The 15 patients included in this cohort represent all operable cases using the described technique, as excluding them would have been contrary to international guidelines. Metastatic disease is considered non-operable, whereas local or locally invasive disease is operable, with the latter requiring the addition of lymph node dissection. We believe including patients with all clinical stages in the cohort strengthens the study's conclusions.
- Why was only 12-month follow-up applied?
Author’s response: For UTUC, urological follow-up is conducted every three months during the first year postoperatively, in parallel with oncology follow-up. The 12-month period was defined as the minimum follow-up duration, although most patients had a longer follow-up period.
We added lines 172-176:
“A period of 12 months was defined as minimum time-frame for follow-up. This period characterized by a close monitoring, thus, the majority of metastatic or recurrence cases are identified within this timeframe. Although international guidelines indicate that close follow-up has limited evidence for both high- and low-risk disease following radical NU, our institution continues to adhere to this practice, ensuring diligent monitoring (Rouprêt et al., 2023). “
- What is the indication for lymphadenectomy, please explain.
Author’s response: According to the guidelines, lymph node dissection (LND) is recommended only for high-risk disease or when suspected lymph nodes are identified intraoperatively. Following the guidelines, we performed LND for high-risk, non metastatic patients, as noted in Line 65.
- Considering the uncertainties mentioned in sub-items 2 and 4, please provide a comprehensive diagnostic protocol, exclusive of URS, detailing the specific tests that should be conducted on each patient deemed eligible for the procedure. (A flow chart may be the most appropriate format for presentation.)
Author’s response: This study focuses on surgical technique rather than oncological decision-making. Until April 2024, a biopsy-proven diagnosis was a mandatory pre-surgical requirement, with ureteroscopy being the preferred method to obtain a biopsy. Cytology was also an option but often insufficient for a definitive diagnosis. Following the publication of the latest guidelines, decisions can now be made based on imaging alone when high-risk disease is suspected. However, these guidelines were implemented after the data collection for this cohort.
This is standard urological knowledge and falls outside the primary focus of our study. Once a patient is diagnosed with non-metastatic UTUC, the gold standard of treatment remains surgical intervention. Consequently, all clinical stages were included in this study. As of April 2024, (Pandolfo et al., 2024; Rouprêt et al., 2023), neoadjuvant therapy was still uncommon, and the data collection predated these guideline updates.
Nonetheless, our study remains highly relevant, as it deals with surgical technique rather than oncological management, which is more pertinent to metastatic disease. Our goal was to provide a detailed and reproducible description of the surgical method that could serve as a practical guide for surgeons looking to adopt this technique in their practice.
Reviewer 2 Report
Comments and Suggestions for Authors
The paper in review is a presentation for an improved technique of radical robotic nephroureterectomy for patients with upper urinary tract cancer. The technique in question involves only one docking of the robot console, the targeting of 2 zones using the camera hop function of the daVinci xi system and the use of only 3 ports.
Introduction was well written and acts as a good preamble to the paper.
The group of patients was well defined and the surgery was explained step by step, from positioning of the patients to the surgical technique , which is helpful especially for surgeons that want to implement it in their center.
Results were clear, all patients follow up was presented, especially the ones that presented complications.
Discussions were in line with the theme of the paper, other papers that treatise on the use of the surgical robotic system for radical neproureterectomy were taken into account.
The authors show that the modifications on the RRNU technique that are presented can offer good postoperative results with a shorter operative time (the authors showed a median operative time of 162 minutes with 60% of surgeries taking less than 150 minutes) and also offering improved logistics in the OR - showing the great experience of the surgeon that performed them.
These types of papers are necessary in order to improve surgical techniques - standardizing operative times and techniques will improve the way patients are managed overall - regardless of the surgeon and place where the surgery takes place.
Surely more research is needed, with the adoption of the technique described in more centers and by more surgeons.
For the reasons stated above and for the role of these papers to bring new techniques to surgeons all around the world, I recommend for publication.
Author Response
Author's Reply to the Review Report (Reviewer 2)
Dear Reviewer II,
Thank you for taking the time to review our manuscript and for providing such thoughtful and encouraging feedback. We appreciate your detailed assessment of our study and the recognition of its potential contribution to the field of urological surgery.
Reviewer 3 Report
Comments and Suggestions for Authors
The authors analyzed the results of 15 patients who underwent robotic radical nephroureterectomy via the one docking, two-target, and three-port technique in the single-center cohort study between 2019 and 2024.
Some comments are listed below.
1. Figure 1 needs a more detailed figure legend. The font in the figure is too small to visualize.
2. Line 37: Correct the error. The mean age was 74.5 years old, and the median age was 79 years old.
3. Line 170: Figure 2 is not included in the current manuscript.
4. All numbers in the Tables should be organized in the same column.
5. Table 3: <150 min is 9 (60%) and ≥150 min is 7 (40%). However, the total number of patients is 15. Is there any error? Provide the full name of EBL at the end of the table.
6. Table 4: Provide the full name of TNM at the table's end.
7. Lines 260-261: [These complications should be considered in light of our patient population, which was mostly octogenarians (Table 1)]. What are the authors' opinions and recommendations for treatment options for octogenarians due to the high mortality rate? Any references?
8. Line 267 should list all references mentioned in the line 265.
9. Line 273: The reference citation format should be consistent.
10. Table 6: Subgroups of Metastatic disease are out of order.
11. Patient consent statement should include in the manuscript.
Author Response
Author's Reply to the Review Report (Reviewer 3)
Dear Reviewer III,
Thank you for your insightful and constructive review. Thanks to your comment we could improve our work. We have followed your notes and made the following changes:Bottom of Form
- Figure 1 needs a more detailed figure legend. The font in the figure is too small to visualize.
Author’s response: Was reviewed and changed, we have added in the figure legend the missing details.
- Line 37: Correct the error. The mean age was 74.5 years old, and the median age was 79 years old.
Author’s response: Thank you for this comment. We have changed the sentence from “the mean age was 79” to the “the median age was 79”, Line 37.
- Line 170: Figure 2 is not included in the current manuscript.
Author’s response: Line 170: “At 12 months, overall survival and disease-free survival rates were 78.57% and 42.86%, respectively (Figure 2.)”->
Line 177-178:
“At 12 months, overall survival and disease-free survival rates were 78.57% and 42.86%, respectively.”
- All numbers in the Tables should be organized in the same column.
Author’s response: We have aligned all number in all tables in the edited manuscript.
- Table 3: <150 min is 9 (60%) and ≥150 min is 7 (40%). However, the total number of patients is 15. Is there any error? Provide the full name of EBL at the end of the table.
Author’s response: Indeed we had a typo, we have corrected the number of >150 min to 6 which was the right data. Line 258.
- Table 4: Provide the full name of TNM at the table's end.
Author’s response: Thank you, we have detailed the TNM abbreviation. Line 276.
- Lines 260-261: [These complications should be considered in light of our patient population, which was mostly octogenarians (Table 1)]. What are the authors' opinions and recommendations for treatment options for octogenarians due to the high mortality rate? Any references?
Author’s response: Thank you for this enlightening note. We have added a short literature review based on our subgroup analysis.
Lines 193-207:
“Sub-group analysis:
With regard to Charlson adjusted comorbidity index (CCI): It is important to note that the cohort's population had a median CCI score of 8. To assess whether this technique is suitable for highly morbid patients, we conducted a sub-analysis using a CCI cut-off point of ≥5. A total of 11 out of the 15 patients met this criterion. Among them, 2 patients died from metastatic disease, 1 died from cardiac arrest as mentioned, and 2 patients are still alive, presenting with metastatic UTUC and receiving adjuvant therapy. The overall survival rate in this sub-analysis group was 72.72%, and the disease-free survival rate was 54.54%.
With regard to age: Regarding age, the cohort's median age was 79 years. We conducted a sub-group analysis to assess whether the elderly population would benefit from the procedure, using 80 years as the cut-off. A total of 7 patients were 80 years or older. Of them, only one patient died from metastatic disease. The remaining 6 patients survived with no evidence of disease at the 12-month follow-up. The overall survival rate in this subgroup was 85.71%, with the same rate of disease-free survival.”
Lines 314-347:
“These complications should be considered in the context of our patient population, which primarily consisted of patients with a high burden of comorbidities, as reflected by a median CCI of 8 (Table 1). The CCI is designed to pre-operatively identify patients at greater risk of intra- and postoperative complications (AlAshqar et al., 2023). In our analysis of the subgroup of patients with a CCI ≥ 5, we observed a total complication rate of 18%.
The inclusion of octogenarian patients in the surgical management of UTUC has been a frequent focus of previous studies. Large cohorts have demonstrated the benefits of minimally invasive RRNU in elderly populations, emphasizing its low complication rates and relatively high overall survival rates (Koterazawa et al., 2023a, 2023b; Teoh et al., 2022; Trecarten et al., 2024).
In our sub-analytical group of patients aged over 80 years, this study corroborates earlier findings, confirming the safety and feasibility of minimally invasive NU. Within this group, we observed a relatively high survival rate of 85.7% and an identical disease remission rate, with a mean follow-up period of 18 months. The complication rate in this subgroup was 14%, comparable to the overall study population.
A review of the literature reveals diverse opinions regarding the factors that most significantly influence surgical outcomes. Wallace et al. reported higher mortality rates among elderly patients; however, their study encompassed 12 different urological procedures rather than focusing solely on NU (Wallace et al., 2018). In contrast, data from 2,352 patients in the CROES-UTUC registry demonstrated survival benefits associated with radical NU, regardless of age or comorbidities (Teoh et al., 2022).
Our findings suggest that a patient’s morbidity profile, as reflected by the CCI, may have a more significant impact on complication rates than age alone (AlAshqar et al., 2023). Nonetheless, further prospective studies are necessary to validate these findings.
It is worth noting, however, that the majority of clinical research supports the superiority of radical NU for UTUC, even in the octogenarian population(Koterazawa et al., 2023b; Teoh et al., 2022; Upfill-Brown et al., 2019; Ye et al., 2023). Multiple clinical trials have compared minimally invasive approaches to open surgery, all indicating the superiority of the minimally invasive approach, particularly robotic surgery (Franco et al., 2023; H. Lee et al., 2019; Rajan et al., 2023; Trecarten et al., 2024; Ye et al., 2023). Unfortunately, one of the drawbacks of the robotic approach, as mentioned, is its time consumption, which may limit its use in the fragile elderly population, further underscoring the advantage of our time-efficient surgical technique (Maynou et al., 2023; Reddy et al., 2023).”
- Line 267 should list all references mentioned in the line 265.
Auther’s response: Changes were made. Lines 350-353
Studies by Nanigian et al. (2006), Lee et al. (2013), Patel et al. (2015), Pathak et al. (2020), and others have reported various robotic approaches; however, they did not employ this specific configuration (Z. Lee et al., 2013; Nanigian et al., 2006; Patel et al., 2015; Pathak et al., 2020)
- Line 273: The reference citation format should be consistent.
Auther’s response: Changes were made.
- Table 6: Subgroups of Metastatic disease are out of order.
Auther’s response: Changes were made.
- Patient consent statement should include in the manuscript.
Auther’s response: All patients have signed an operative informed consent. The institutional ethics committee approved this study with an exemption from any additional informed consent, as the surgical management is a common practice, and a variety of techniques can be employed and tailored to each case according to surgeon expertise. The detailed Helsinki approval was shared with the Journals’ editor.
Round 2
Reviewer 1 Report
Comments and Suggestions for Authors
I acknowledge the authors' adequate response to comments 1-4.
However, I respectfully disagree with the assertions that "This is standard urological knowledge and falls outside the primary focus of our study" and "This study focuses on surgical technique rather than oncological decision-making".
This scientific article should clearly present the surgical technique while also addressing the oncological context. As the authors have correctly noted, guidelines evolve; therefore, a clear diagnostic schema should be provided both for current and future reference, as without it, the evaluation of surgical treatment outcomes becomes unfeasible.
Author Response
Dear Reviewer I,
We sincerely appreciate your thoughtful feedback and acknowledge the importance of providing a clear oncological context alongside the description of our surgical technique. While our primary focus is on the surgical approach, we agree that outlining a structured diagnostic schema is essential for contextualizing our findings and ensuring the reproducibility of our results, particularly as guidelines continue to evolve.
To address this, we have revised the manuscript to include a more detailed diagnostic framework, clarifying the preoperative evaluation process and decision-making criteria leading to radical nephroureterectomy (NU). We believe this addition enhances the clarity and applicability of our study for both current and future clinical practice.
We hope this revision aligns with your concerns and appreciate your valuable insights in strengthening our manuscript. The following changes have been made:
* lines 82-87:
"All patients included in the study had a suspicious lesion on imaging and were diag-nosed with UTUC through diagnostic ureteroscopy (Dx URS). Following a multidisciplinary evaluation of all associated risks, they were referred for radical NU (Scheme 1). Notably, updated guidelines now allow a radical surgical approach based solely on imaging or visual appearance, without requiring biopsy-confirmed malignancy(Pandolfo et al., 2024). However, our study was conducted before these guideline revisions. "
* We added Scheme1: management of UTUC utilized in our cohort (lines 90-112)
Reviewer 3 Report
Comments and Suggestions for Authors
Line 117: Line number 97 is on the top of Figure 1. The words and icon in the bottom of Figure 1 could be larger to visualize. Figure 1 legend: The space between words need be corrected.
Author Response
Dear Reviewer,
We appreciate your quick review and attention to the small and important details. We have made the changes in the Figure 1 legend and spaced between lines 97 to 117.
Round 3
Reviewer 1 Report
Comments and Suggestions for Authors
Thanks to authors to revised manuscript according my suggestions.